# Towards Foundation Models for Quantum Unitary Synthesis via Zero-Shot MDL

**Lukas Theißinger**
University of Bonn
Lamarr Institute
`lukas.theissinger@uni-bonn.de`

**Thore Gerlach**
European Space Agency (ESA)

**David Bergauhs**
Fraunhofer IAIS
Lamarr Institute

**Christian Bauckhage**
University of Bonn
Lamarr Institute
Fraunhofer IAIS

## Abstract

Quantum unitary synthesis addresses the problem of translating abstract quantum algorithms into sequences of hardware-executable quantum gates. Solving this task exactly is infeasible in general due to the exponential growth of the underlying combinatorial search space. Existing approaches suffer from misaligned optimization objectives, substantial training costs and limited generalization across different qubit counts. We mitigate these limitations by using supervised learning to approximate the minimum description length of residual unitaries and combining this estimate with stochastic beam search to identify near optimal gate sequences. Our method relies on a lightweight model with zero-shot generalization, substantially reducing training overhead compared to prior baselines. Across multiple benchmarks, we achieve faster wall-clock synthesis times while exceeding state-of-the-art methods in terms of success rate for complex circuits.

## 1 Introduction

Quantum computing is compelling in principle because quantum algorithms exploit superposition and interference to implement transformations that appear qualitatively inaccessible to classical computation (Nielsen & Chuang, 2010; Montanaro, 2016; Shor, 1994; Grover, 1996). In practice, those algorithmic advantages are realized through unitary operations: a quantum algorithm is, at its core, a sequence of unitary matrices. This makes a simple but often under-emphasized point unavoidable: progress in quantum algorithms is bottlenecked not only by hardware, but by our ability to construct circuits that implement the right unitaries with realistic resource costs (Nielsen & Chuang, 2010). The task of finding a sequence of gates to build a quantum circuit which realizes a given unitary is called Quantum Unitary Synthesis (QUS) (Shende et al., 2005).

QUS closely parallels symbolic regression (Augusto & Barbosa, 2000). In both cases, the target is observed only through numerical representations (i.e., matrix elements of a unitary or floating-point evaluations of a function) while the objective is to recover a compact description composed from a finite set of primitives. This task goes beyond numerical approximation: distinct symbolic structures may induce similar numerical behavior, whereas small symbolic errors can lead to large numerical discrepancies. This suggests that QUS, like symbolic regression, is fundamentally a search over discrete symbolic spaces rather than continuous optimization in isolation.

Classical approaches to QUS rely on heuristic search (Paradis et al., 2024), exact optimization (Nagarajan & Zsolt, 2025) or hand-crafted algebraic rules (Gheorghiu et al., 2022), but remain effective only in limited regimes. As system size increases, synthesis is hindered by the combinatorial explosion of possible gate sequences. Commonly used numerical objectives provide only weak guidance: numerical proximity between unitaries does not necessarily reflect symbolic similarity. Locally optimal choices may block globally correct constructions. State-of-the-art supervised learning methods (Fürrutter et al., 2024; Barta et al., 2025; Chen & Tang, 2025) suffer from similar limitations.

To address this mismatch, Yu et al. (2025) propose using the Minimum Description Length (MDL) (Kolmogorov, 1963) as a structurally meaningful objective. It is defined as the smallest number of symbols required to represent an expression. The MDL decreases monotonically along correct symbolic sequences and thus provides guidance for exploration in discrete symbolic spaces, in opposition to conventional norm-based distance metrics. Recent methods based on reinforcement learning (RL) attempt to utilize this by designing reward functions to reflect symbolic similarity (Rietsch et al., 2024; Kremer et al., 2025). However, these models require long training times and exhibit limited generalization across different qubit counts.

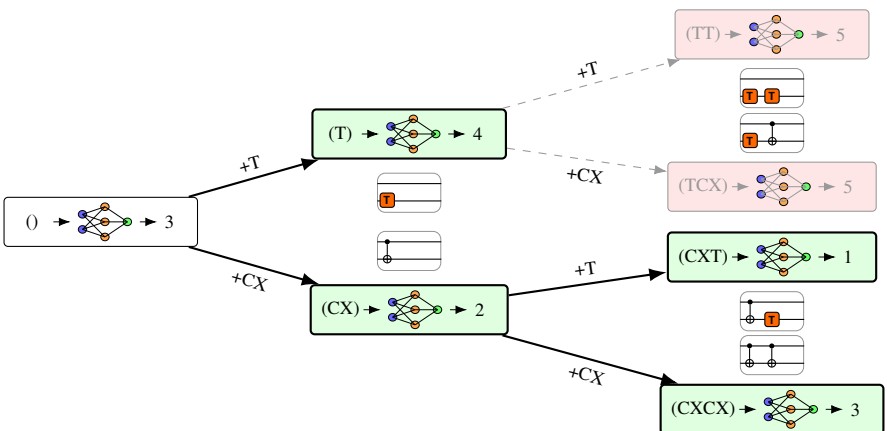

Figure 1: **Fast and Scalable Quantum Circuit Synthesis** Illustration of our synthesis search. We use an MDL predictor with beam search at inference time: candidate circuits are expanded by appending gates (+T, +CX). Each node denotes a partial gate sequence (T, CX, TCX, CXT, TT, CXCX) and the numbers show the predicted remaining MDL. Green nodes are kept, while red nodes are pruned from the search.

In this work, we propose an RL-free approach to QUS that leverages the efficiency of supervised learning while adapting the MDL framework to overcome the mismatch between symbolic structure and reconstruction error. We generate synthetic training data by sampling random quantum circuits and their corresponding unitaries, enabling supervised training to predict the MDL of candidate partial circuits. The trained model is then used as a value function within a beam-search procedure to efficiently explore the combinatorial space of circuit decompositions. An overview of our method is found in Fig. 1 and our contributions are summarized as follows:

- **Synthesis via MDL**: We formulate quantum circuit synthesis as the problem of estimating the remaining optimal gate cost of a residual target unitary using the MDL, yielding a structurally meaningful value function for guiding symbolic search.

- **Lightweight Model**: We find that a lightweight multi-layer perceptron achieves accuracy better than a transformer architecture (Vaswani et al., 2017) which one might expect to be more expressive (Sec. A), resulting in fast inference. Training time is largely reduced compared to state-of-the-art RL approaches (Rietsch et al., 2024; Kremer et al., 2025).

- **Zero-shot Capabilities**: We train a single model on a broad synthetic distribution and deploy it *zero-shot*, without task-specific retraining, across all evaluation settings. Consistent with prior findings (Seifner et al., 2025; Berghaus et al., 2025), this enables effective generalization, including to circuits with varying numbers of qubits, while avoiding the costly per-qubit training required by previous approaches.

- **State-of-the-Art Performance**: We show that our method is state-of-the-art in both time and recovery rate by comparing it against previous classical and RL-based baselines on a variety of established benchmarks (Lu et al., 2023).

## 2 RELATED WORK

A large body of work focuses on classical optimization for quantum circuit synthesis with arbitrary gate sets, exploiting the maturity of general-purpose solvers. `Synthetiq` (Paradis et al., 2024) utilizes simulated annealing (Kirkpatrick et al., 1983) to iteratively refine a randomly built circuit until a certain criterion is met. `QuantumCircuitOpt` (Nagarajan et al., 2021; Nagarajan & Zsolt, 2025) converts the exact circuit problem into a mixed-integer linear program, enabling the use of sophisticated branch-and-bound solvers such as `Gurobi` (Gurobi Optimization, LLC, 2024). Automated reasoning is used in Zak et al. (2025) by casting unitary synthesis as a model counting problem. While these approaches provide strong optimality guarantees and broad expressivity, their reliance on general-purpose solvers often limits scalability as circuit size and search space grow.

For Clifford+T synthesis, strong guarantees are provided by exploiting the algebraic structure of the gate set. Kliuchnikov et al. (2013) showed that single-qubit unitaries admit exact, ancilla-free decompositions and described an efficient construction strategy for recovering the corresponding circuit. This result was extended to the multi-qubit regime, presenting constructive algorithms that match theoretical guarantees (Giles & Selinger, 2013; Gheorghiu et al., 2022). A more efficient heuristic for synthesizing a T-count optimal unitary was developed in Mosca & Mukhopadhyay (2021). However, these algorithms exhibit exponential runtime in the worst case, restricting their practical use to small instances. The underlying number-theoretic structure nevertheless motivates our focus on gate count-aware search over the same gate family.

Recent progress in unitary synthesis complements exact methods with data-driven heuristics, reflecting a broader shift away from purely symbolic and heuristic techniques (Wang et al., 2022). The most prominent supervised methods rely on generative diffusion models (Fürrutter et al., 2024; Chen & Tang, 2025; Barta et al., 2025).

QAS-Bench further systematizes this perspective by providing a reusable benchmark suite that exposes dozens of parameterized search tasks, enabling controlled comparisons between competing heuristics (Lu et al., 2023). Our beam-search blueprint borrows the benchmarking philosophy (fixed search budgets, standardized action sets) and adapts it to the Clifford+T domain.

Owing to the vast design space of quantum circuits, several recent works have investigated the automation of unitary synthesis using reinforcement learning (RL) (Kundu et al., 2024; Kremer et al., 2024; Zen et al., 2025; Kundu & Mangini, 2025; Rietsch et al., 2024). Building on the tree-search–based AlphaZero algorithm (Silver et al., 2018), these approaches learn value functions that guide the search toward promising unitary synthesis strategies (Rietsch et al., 2024). These methods have been extended to dynamic circuits (Valcarce et al., 2025) and the Pauli channel representation (Kremer et al., 2025). While these methods demonstrate substantial improvements over unguided search, they remain restricted to a fixed number of qubits. We pursue a complementary direction in which a compact neural network produces gate count estimates that bias stochastic beam expansions, yielding a simple yet effective heuristic.

## 3 BACKGROUND

We give a brief primer on circuit-model quantum computing and *quantum unitary synthesis* (QUS); see Nielsen & Chuang (2010) for a full introduction. We also cast QUS as an MDP to motivate our search formulation.

**Quantum Computing** An $n$-qubit state $|\psi\rangle$ is a unit vector in $\mathbb{C}^{2^n}$. Computation applies unitary matrices (*quantum gates*); a quantum circuit is a sequence of such gates, optionally followed by measurement to produce classical outcomes. This representation makes the exponential state space explicit and is the basis for circuit synthesis.

**Quantum Unitary Synthesis** In the circuit model, a circuit $C = (\mathbf{G}_1, \ldots, \mathbf{G}_m)$ with $\mathbf{G}_i \in \mathbb{C}^{2^n \times 2^n}$ implements $\mathbf{U}(C) = \mathbf{G}_m \cdots \mathbf{G}_1$. QUS seeks a circuit that implements a target unitary $\mathbf{U}^\star$ under gate-set constraints:

$$\underset{C=(\mathbf{G}_1, \ldots, \mathbf{G}_m)}{\arg\min} d\left(\mathbf{U}(C), \mathbf{U}^\star\right), \quad \mathbf{G}_i \in \mathcal{G},$$

where $d$ measures unitary mismatch and $\mathcal{G}$ is the allowed gate set. Common choices for $d$ include worst-case error, Hilbert-Schmidt distance, and average gate fidelity. We use the average gate fidelity

$$F_{\text{avg}}(U, V) = \frac{|\operatorname{Tr}(U^\dagger V)|^2 + D}{D(D + 1)}, \qquad D = 2^n, \tag{1}$$

which has a direct operational interpretation (Nielsen, 2002; Gilchrist et al., 2005).

**Considered Gate Set** We use the Clifford+T family, a universal gate set common in fault-tolerant architectures that induces a discrete search space (Bravyi & Kitaev, 2005; Kliuchnikov et al., 2013; Amy et al., 2013).

Intuitively, synthesis can be viewed as repeatedly explaining away the target unitary: after fixing a partial circuit, the remaining task is whatever transformation has not yet been accounted for. This perspective makes it natural to search over *residual* unitaries rather than complete circuits and motivates the state definition below.

**Decision Process Formulation** QUS can be expressed as an MDP over residual unitaries. Given a partial circuit prefix $C_{1:t}$ with unitary $\mathbf{U}_{1:t} = \mathbf{U}(C_{1:t})$, define

$$\mathbf{R}_t = \mathbf{U}_{1:t}^\dagger \mathbf{U}^\star, \tag{2}$$

and let an action apply a gate $\mathbf{G} \in \mathcal{G}$, yielding $\mathbf{R}_{t+1} = \mathbf{G}^\dagger \mathbf{R}_t$. We use the reward

$$r(\mathbf{R}_t, \mathbf{G}) = \begin{cases} 0, & d(\mathbf{R}_t, \mathbf{I}) \leq \epsilon, \\ -1, & \text{otherwise.} \end{cases}$$

Under this reward, the optimal value equals the negative remaining gate count. The resulting search space is exponential, so we rely on learned heuristics to guide exploration rather than explicit graph expansion.

## 4 METHODOLOGY

To connect synthesis to MDL, we view a circuit as a discrete description string over the alphabet $\mathcal{G}$. Let $\ell(\mathbf{G}) \geq 0$ denote the (fixed) expression-length assigned to gate $\mathbf{G}$ under a chosen encoding and define the description length of a circuit as

$$\ell(C) \triangleq \sum_{i \leq m} \ell(\mathbf{G}_i).$$

The MDL of a unitary under $\mathcal{G}$ is the length of its shortest description

$$\text{MDL}_\mathcal{G}(\mathbf{U}) \triangleq \min_{C: \, d(\mathbf{U}(C), \mathbf{U}) \leq \epsilon} \ell(C).$$

We choose the simple instantiation $\ell(\mathbf{G}) \equiv 1$, from which follows $\ell(C) = |C|$ and $\text{MDL}_\mathcal{G}(\mathbf{U})$ is exactly the minimum gate count needed to represent $\mathbf{U}$ over $\mathcal{G}$.

A convenient state representation for guided search is the residual unitary induced by a partial circuit as defined by (Eq. (2)). Completing the synthesis from the current prefix is equivalent to exactly synthesizing $\mathbf{R}_t$ and the remaining optimal future cumulative reward is precisely the MDL. In other words, $\text{MDL}_\mathcal{G}(\mathbf{R}_t)$ is an ideal cost-to-go: a perfect heuristic would predict how many additional gates are required from the current search state. Our method operationalizes this MDL view by learning to predict $\text{MDL}_\mathcal{G}(\mathbf{R}_t)$ and using that prediction to prioritize expansions in the search.

### 4.1 PREDICTING THE MINIMUM DESCRIPTION LENGTH

We train the MDL predictor in a supervised fashion, which requires pairs of residual unitaries and their remaining MDL. Since computing the exact $\text{MDL}_\mathcal{G}(\cdot)$ is computationally hard, we generate *consistent* labels from heuristically optimized circuits and treat the resulting targets as an approximation to MDL that is sufficient for guiding the search.

**Data Generation** We construct training circuits via rejection sampling to control circuit difficulty and keep the marginal distribution over target T-count approximately uniform. Concretely, we draw

a target T-count $k \in [0, 20]$ uniformly at random and propose a random Clifford+T circuit by (i) sampling a random Clifford circuit, choosing Clifford gates and their acted-on qubits uniformly at random, (ii) inserting exactly $k$ T-gates at uniformly sampled positions and (iii) randomly shuffling the resulting gate sequence. We then apply a lightweight peephole optimizer (a fixed set of local rewrite rules) to remove obvious cancellations and normalize the circuit; if the optimizer changes the T-count, we reject the proposal and resample.

We use a lightweight curriculum to better train on high T-count circuits. Let $C$ be an accepted optimized circuit with unitary $\mathbf{U}^\star = \mathbf{U}(C)$ and gate count $|C|$. We form multiple supervised examples by choosing cut positions $t \in \{0, \ldots, |C|\}$ and taking the prefix $C_{1:t}$ with unitary $\mathbf{U}_{1:t} = \mathbf{U}(C_{1:t})$. The search state is the residual unitary

$$\mathbf{R}_t \;=\; \mathbf{U}_{1:t}^\dagger \mathbf{U}^\star,$$

i.e., the unitary remaining after committing to the prefix. The ideal target is then $\mathrm{MDL}_{\mathcal{G}}(\mathbf{R}_t)$. Since $C_{t+1:|C|}$ implements $\mathbf{R}_t$, we optionally re-run the peephole optimizer on this suffix to obtain a shorter exact description $\widetilde{C}_t$ and set the label $y_t = |\widetilde{C}_t|$. Under $\ell(\mathbf{G}) \equiv 1$, $y_t$ proxies the remaining minimum gate count.

To emphasize hard (long-residual) states, we do not sample $t$ uniformly. Instead, we align cuts with non-Clifford structure: letting $k'$ be the number of $T$ gates in $C$, if $k' \geq 5$ we include a cut after the $\lfloor k'/2 \rfloor$-th $T$ gate, and if $k' \geq 10$ we additionally cut after the $\lfloor 3k'/4 \rfloor$-th $T$ gate (in circuit order). This broadens the label distribution and upweights high-$T$ circuits.

**Model Input** Each residual unitary $\mathbf{R}_t$ is converted into the network input by removing an arbitrary global phase and stacking real and imaginary parts into a real-valued tensor (we discuss alternative tokenization methods in appendix A).

We remove the arbitrary global phase by aligning the phase of the first non-negligible (row-major) entry to be real and nonnegative, yielding a phase-invariant representation. This is necessary because global phase is physically unobservable.

**Loss Function** Given a phase-normalized residual $\widehat{\mathbf{R}}_t$, the network $f_\theta$ produces a single scalar prediction $\hat{y}_t = f_\theta(\widehat{\mathbf{R}}_t)$, intended to estimate the remaining description length $y_t = |\widetilde{C}_t|$. We train $f_\theta$ by standard squared-error regression:

$$\mathcal{L}(\theta) \;=\; \mathbb{E}_{(\widehat{\mathbf{R}}_t, \, y_t) \sim \mathcal{D}}\big[\big(f_\theta(\widehat{\mathbf{R}}_t) - y_t\big)^2\big],$$

implemented as the mean squared error over minibatches. At test time, $-\hat{y}_t$ is used as an estimate of the value function.

## 4.2 Inference with Stochastic Beam Search

At inference time we are given a target unitary $\mathbf{U}^\star$ and a trained predictor $f_\theta(\cdot)$ that estimates the MDL, i.e., the remaining minimum gate count. If $f_\theta$ were perfect, repeatedly choosing the next gate that minimizes the predicted remaining MDL would recover an optimal circuit. In practice the predictor is imperfect (both due to approximation error and because our labels are proxy MDL values), so purely greedy decoding is brittle. We therefore use a width-bounded search that mixes exploitation with controlled exploration while retaining high throughput.

**Approximate Optimal Value Function** To rank candidates during beam search, we approximate the optimal value function by the negative predicted MDL:

$$V^*(\mathbf{R}_t) \approx -f_\theta(\mathbf{R}_t).$$

Beam search keeps the top $B$ residuals under this score, expands each one by all allowed gate applications, evaluates $f_\theta$ on the resulting residuals in a single batch and selects the top $B$ successors for the next beam.

**Stochastic Selection** To avoid over-committing to the model early, we use a stochastic selection rule based on Gumbel-top-$B$ sampling (Danihelka et al., 2022): we add i.i.d. Gumbel noise to the negative scores and take the top-$B$. This is equivalent to sampling without replacement from a softmax distribution over $-f_\theta$ with a softness-controlling temperature $\tau$:

$$\arg\mathrm{top}_{\mathbf{G}} \left( -f_\theta(\mathbf{G}^\dagger \mathbf{R})/\tau + g, B \right), \quad g \sim \mathrm{Gumbel}(0, 1).$$

---

**Algorithm 1** Stochastic Beam Search Inference

---

**Require:** Target unitary $\mathbf{U}^\star$, gate set $\mathcal{G}$, MDL predictor $f_\theta$, beam width $B$, max steps $T$, temperature $\tau$, goal
    tolerance $\varepsilon$
**Ensure:** A circuit $C$
 1: $\mathbf{R}_0 \leftarrow \mathbf{U}^\star$;   beam $\leftarrow \{(\mathbf{R}_0, \langle\rangle)\}$;   sol $\leftarrow \emptyset$
 2: **for** $t = 0$ to $T - 1$ **do**
 3:     cand $\leftarrow \emptyset$
 4:     **for all** $(\mathbf{R}_t, C_t)$ in beam **do**
 5:        **if** $d(\mathbf{R}_t, \mathbf{I}) \leq \varepsilon$ **then**
 6:           sol $\leftarrow$ sol $\cup \{(C_t, t)\}$
 7:           **continue**
 8:        **end if**
 9:        **for all** $\mathbf{G} \in \mathcal{G}$ **do**
10:           $\mathbf{R}_{t+1} \leftarrow \mathbf{G}^\dagger \mathbf{R}_t$
11:           $C_{t+1} \leftarrow C_t \parallel \mathbf{G}$
12:           mdl $\leftarrow -f_\theta(\mathbf{R}_{t+1})$
13:           cand $\leftarrow$ cand $\cup \{(\mathbf{R}_{t+1}, C_{t+1}, \text{mdl})\}$
14:        **end for**
15:     **end for**
16:     **if** sol $\neq \emptyset$ **then**
17:        **return** best $C$ in sol
18:     **end if**
19:     **if** cand $= \emptyset$ **then**
20:        **break**
21:     **end if**
22:     beam $\leftarrow$ Top-$B\big(\text{cand}, \ \text{mdl}/\tau + \text{Gumbel}(0, 1)\big)$
23: **end for**

---

This yields a simple exploration-exploitation tradeoff with minimal engineering overhead and preserves the key advantage of beam search in our setting: the entire expansion-and-scoring step is completely parallel across candidates.

**Termination Criterion** We declare convergence once the iterate is $\varepsilon$-close to the identity under a phase-invariant notion of discrepancy, i.e., $d(\mathbf{R}_t, \mathbf{I}) \leq \varepsilon$. We measure this closeness using the average gate fidelity (Nielsen, 2002), defined in Eq. (1).

Finally, we run multiple independent trials and keep the best solution. Each trial corresponds to an independent stochastic run (different Gumbel noise) and can also apply symmetry transformations that preserve synthesis difficulty but change the search landscape. In particular, permuting qubits via a permutation matrix $\mathbf{P}$ yields the equivalent target $\mathbf{U}_\pi^\star = \mathbf{P}\mathbf{U}^\star\mathbf{P}^\dagger$; a circuit for $\mathbf{U}_\pi^\star$ is converted back by relabeling the corresponding qubit indices. When the action set is closed under adjoints, we additionally allow inverse trials that synthesize $\mathbf{U}_\pi^{\star\,\dagger}$ and convert the result to a circuit for $\mathbf{U}^\star$ by reversing the gate sequence and taking adjoints. In all cases, trials are independent and parallelizable. A pseudo code of our inference method can be found in Alg. 1.

## 5   EXPERIMENTS

We trained one MDL predictor on $n = 5$ qubits once for all further experiments. The MDL-predictor is a feed-forward multilayer perceptron with hidden dimensions $(1024, 512, 128)$ and a softplus output to ensure non-negative predictions. Training the MLP takes approximately 6 hours in our implementation and is dominated by data generation and unitary construction; we use 30 CPU cores and a 4 GB GPU. This is substantially cheaper than the RL training regime reported in Rietsch et al. (2024) (7 days of training). We also trained a transformer model with different tokenization strategies, which did not yield any improvement over the MLP (see Sec. A).

To apply the MDL predictor to a different number $m < 5$ of qubits, we embed an $m$-qubit target unitary $\mathbf{U} \in \mathbb{C}^{2^m \times 2^m}$ into the 5-qubit space by padding with identity,

$$\mathbf{U}_{\text{pad}} \ = \ \mathbf{U} \otimes \mathbf{I}_{2^{5-m}}, \tag{3}$$

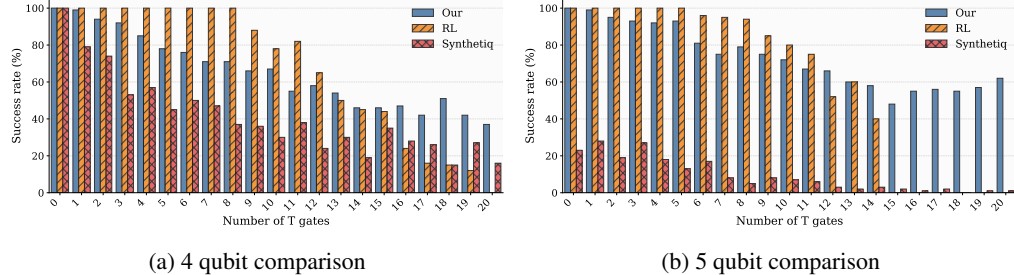

(a) 4 qubit comparison          (b) 5 qubit comparison

Figure 2: **Baseline Comparison**. Success counts (out of 100 targets per T-count, higher is better) for MDL-guided beam search versus the RL baseline of Rietsch et al. (2024) and annealing algorithm of Paradis et al. (2024) on 4 and 5-qubit instances. Our method uses beam width $B{=}10$ and 8000 trials per instance (avg. $\sim$22s runtime) and declares success when $F_{\text{avg}}(\mathbf{U}(C), \mathbf{U}^\star) \geq 0.9$ (Eq. (1)). Results for RL at high T-counts are unavailable because they are not reported in Rietsch et al. (2024), likely due to the associated computational cost.

which corresponds to acting trivially on the additional qubits. Notably, this simple embedding works despite the fact that the model is never trained on padded examples. We emphasize here again that prior learned approaches typically train separate models for each qubit count, while we train only a single model.

## 5.1 EVALUATION ON SYNTHETIC DATA

We evaluate the MDL-guided synthesis on randomly generated Clifford+T targets drawn using the rejection-sampling method described in Sec. 4.1, following the protocol of Rietsch et al. (2024) for comparability: we sample a target T-count and generate a random Clifford+T circuit whose peephole-optimized form preserves that T-count. Unless stated otherwise, we sweep T-counts from 0 to 20 and consider circuits with gate counts in the range $[3, 60]$. For each T-count we generate 100 test instances and report the number of successful syntheses under a fixed compute budget.

We verify synthesis correctness using the phase-invariant average gate fidelity $F_{\text{avg}}$ in Eq. (1). We count a run as successful if the synthesized circuit unitary $\mathbf{U}(C)$ satisfies $F_{\text{avg}}(\mathbf{U}(C), \mathbf{U}^\star) \geq 0.9$. We choose 0.9 for comparability as this is the threshold used in Rietsch et al. (2024). Unless stated otherwise, we use beam width $B{=}10$ and a budget of 8000 trials (Sec. D) per instance. This results in an average runtime of $\sim$22 seconds per instance on an NVIDIA A100-80GB GPU.

Fig. 2 reports synthesis success rate for different numbers of T-count for 4 and 5-qubit instances comparing our approach to RL (Rietsch et al., 2024) and a simulated annealing algorithm `Synthetiq` (Paradis et al., 2024). All methods saturate on low T targets, but performance diverges sharply as T-count increases. In both settings, our approach degrades substantially more slowly: while the RL baseline achieves high success rates for low-to-mid T regime, it drops off rapidly. The main reason for this is most likely as stated in Rietsch et al. (2024)—the training of high T regime is underrepresented during the training process. That is because training the RL model has very high computational cost.

While `Synthetiq` fails on most higher T-count instances, our method maintains a high success rate deep into the hard regime. The gap widens further on 5 qubits, indicating that our approach is substantially more scalable than `Synthetiq`. Since our method takes $\sim$ 22 seconds, we gave `Synthetiq` a timeout of 30 seconds.

**Baseline Note** The comparison between our approach and `Synthetiq` is fair, but we could not re-run the RL baselines from Rietsch et al. (2024) because no code was released, so we rely on the RL success counts reported in their paper; the RL success rates in Fig. 2 should therefore be treated as approximations. Moreover, they did not report per-instance runtime, so we consider the comparison indicative and focus the remaining experiments on controlled ablations and compute-matched baselines that we can run directly.

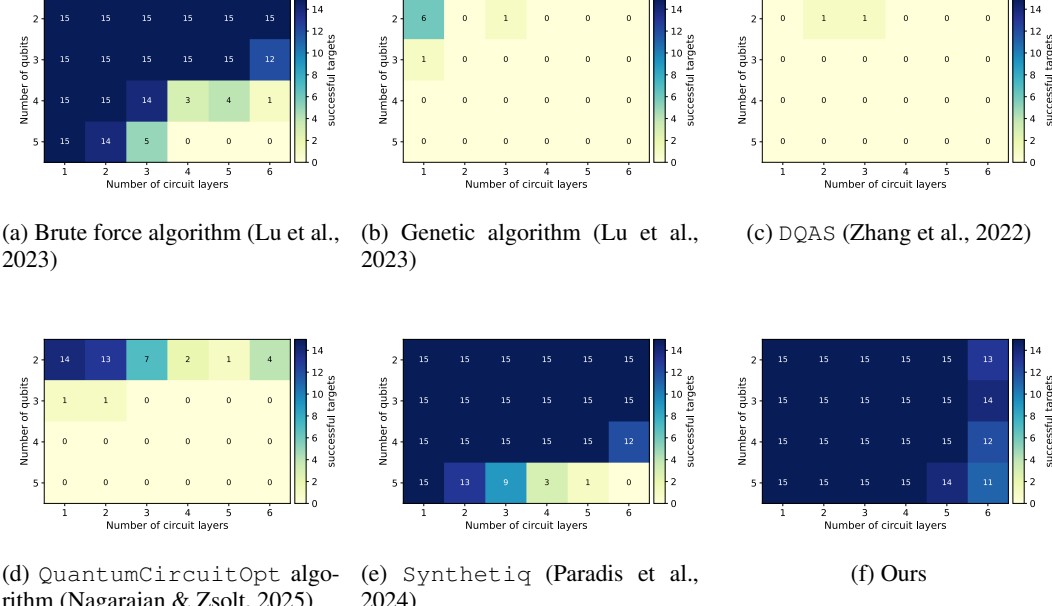

(a) Brute force algorithm (Lu et al., 2023)

(b) Genetic algorithm (Lu et al., 2023)

(c) DQAS (Zhang et al., 2022)

(d) QuantumCircuitOpt algorithm (Nagarajan & Zsolt, 2025)

(e) Synthetiq (Paradis et al., 2024)

(f) Ours

Figure 3: **Standardized Benchmark Comparison**. QAS-Bench (Lu et al., 2023) QC Regeneration results as heatmaps for six methods. Columns correspond to layer difficulty (1–6) and rows to qubit count (2–5). Each cell shows the number of successful syntheses out of 15 targets (5 RC-S + 10 RC-C) for that $(n, \text{layer})$ bucket. All methods are re-run on the same targets under a budget-controlled per-instance wall-clock budget (22 s for ours, 30 s for brute force and Synthetiq, 60 s for all others); darker cells indicate higher success. Our method uses a single $n{=}5$ MDL predictor with padding for $n < 5$, beam width $B{=}10$ and the success criterion $F_{\text{avg}}(\mathbf{U}(C), \mathbf{U}^\star) \geq 0.99$.

## 5.2 Standardized Evaluation on QAS-Bench

To move beyond our synthetic data generation and enable standardized comparison, we evaluate on the QC Regeneration (circuit resynthesis) benchmark in QAS-Bench (Lu et al., 2023). Each instance specifies a target unitary and a candidate gate set; the goal is to recover an equivalent circuit over the same gate alphabet. QAS-Bench provides two folds with closely related gate sets: RandomCircuit-Single (RC-S) uses $\{H, S, T, I\}$, while RandomCircuit-Clifford (RC-C) uses $\{H, S, T, I, \text{CNOT}\}$ (Lu et al., 2023). Both align with our Clifford+T action set (up to the identity), with RC-S forming a strict subset that removes entangling operations.

We evaluate our model across QAS-Bench instances with $n \in \{2, 3, 4, 5\}$, again using the same padding scheme as before (Eq. (3)). QC Regeneration is organized by qubit count and circuit layer depth from 1 to 6. For each $(n, \text{layer})$ bucket, QAS-Bench includes 5 RC-S and 10 RC-C targets (15 total) (Lu et al., 2023). Because this yields only $6 \times 15 = 90$ targets per qubit count, we aggregate RC-S and RC-C and report success counts per $(n, \text{layer})$ bucket. This aggregation is conservative for our method: on RC-S targets we still allow CNOT actions, enlarging the search space rather than simplifying the task. We count a synthesis as successful under the stricter criterion: $F_{\text{avg}}(\mathbf{U}(C), \mathbf{U}^\star) \geq 0.99$.

We follow QAS-Bench's heatmap visualization for direct comparability (Fig. 3). For baselines, we re-run the QAS-Bench reference implementations (bi-directional brute force search and genetic programming) (Lu et al., 2023) and additionally include Synthetiq (Paradis et al., 2024), a differentiable quantum architecture search (DQAS)-style baseline that relaxes discrete gate choices into continuous weights over our gate alphabet and optimizes architecture weights by gradient descent (Zhang et al., 2022) and a provably optimal solver QuantumCircuitOpt that casts circuit synthesis as a discrete mathematical optimization problem (mixed-integer programming) and returns solutions with optimality guarantees when feasible (Nagarajan & Zsolt, 2025).

To keep comparisons budget-controlled to our per-instance runtime ($\approx$22 s), we use wall-clock budgets of 30 s per instance for bi-directional brute force and `Synthetiq` and 60 s for genetic programming, `DQAS` and `QuantumCircuitOpt` to account for higher per-iteration overhead.

As shown in Fig. 3, classical baselines degrade quickly with problem size: bi-directional brute force and `Synthetiq` solve most small instances but fail in the harder multi-qubit settings (notably $n \in \{4, 5\}$ at layers $\geq 4$), while genetic programming and our `DQAS`-style baseline solve only a small fraction in the 2-qubit regime. In contrast, our MDL-guided beam search achieves 15/15 successes in almost every $(n, \text{layer})$ bucket, despite using a single predictor trained on $n{=}5$ and applied to smaller $n$ via padding.

Table 1: Runtime and solution size (lower is better) comparison on common structured circuits. [†]OURS uses a fixed compute budget of 8000 trials per instance (reported time is 22 s for that budget). For other methods, time is wall-clock to produce the reported solution. TIMEOUT indicates exceeding the time limit (1 h). The circuits are explicitly given in Sec. C.

| Problem | OURS budget (s)[†] | gates | SYNTHETIQ time (s) | gates | BRUTE FORCE time (s) | gates | QUANTUMCIRCUITOPT time (s) | gates |
|---|---|---|---|---|---|---|---|---|
| 4-GHZ state | 22 | **4** | 1.10 | 21 | **0.09** | **4** | | TIMEOUT |
| 4-linear cluster state | 22 | **7** | 0.93 | 20 | 3.59 | **7** | | TIMEOUT |
| 4-phase gadget | 22 | **7** | **1.22** | 32 | 28.26 | **7** | | TIMEOUT |
| 5-GHZ state | 22 | **5** | 3.82 | 28 | **1.05** | **5** | | TIMEOUT |
| 5-linear cluster state | **22** | **9** | 132.49 | 23 | 331.26 | **9** | | TIMEOUT |
| 5-phase gadget | **22** | **9** | 224.47 | 49 | $> 742.02$ | — | | TIMEOUT |
| $[[5, 1, 3]]$-perfect code | **22** | **14** | 77.23 | 37 | $> 561.81$ | — | | TIMEOUT |

To further analyze speed and solution quality of our method, we compare on known and common 4 and 5 qubit circuits in Tab. 1. Our approach consistently returns the smallest circuits and where brute force is feasible, matches its optimum. `Synthetiq` produces substantially larger solutions (20-32 gates) despite running in $\approx$1 s. Unfortunately, it cannot directly incorporate the gate minimization task into its objective function, hence it finds solutions very fast that are sub-optimal. At 5 qubits, brute force becomes prohibitively expensive, whereas ours maintains the same 22 s budget and yields compact circuits. `Synthetiq` is both slower and less efficient on the harder tasks, again showing that its scaling is limited. `QuantumCircuitOpt` fails to solve any instance within the 1 h limit (all TIMEOUT).

## 6   LIMITATIONS

**Scope and scalability** All exact-synthesis methods we compare against—including ours—operate on *dense* $n$-qubit unitaries and search states represented explicitly as matrices in $\mathbb{C}^{2^n \times 2^n}$. This choice incurs an unavoidable representation cost: storing a single state already requires $\Theta(4^n)$ complex numbers, and any state expansion that reads/writes full matrices inherits $\Theta(4^n)$ memory traffic and arithmetic up to constant factors. Consequently, the practical $n$ range of *any* dense-matrix exact-synthesis pipeline is limited, independent of whether the heuristic is learned or solver-based. We therefore evaluate up to $n \leq 5$, matching the largest regime where the strongest prior learned and exact baselines report results (or run reliably) under comparable compute and memory budgets. Within this shared dense-matrix setting, our focus is not to change the exponential dependence on $n$, but to improve efficiency *at fixed $n$* by reducing avoidable overhead.

**Other limitations** As with other learned heuristics, performance depends on the training distribution and can degrade on targets that are far outside it; beam search introduces a tunable compute–quality trade-off; and the method does not provide worst-case optimality guarantees.

## 7   CONCLUSION

We introduced a learning-based approach to quantum unitary synthesis that uses a lightweight model to approximate the minimum description length (MDL) of candidate Clifford+T circuits and guides

synthesis via stochastic beam search. The estimator is trained with supervised learning on synthesis data and is reusable across instances, enabling zero-shot inference at test time without per-instance fine-tuning or additional environment interaction. Our results show that a single trained model provides a strong, reusable heuristic: on challenging targets with high T-count, it achieves higher success rates than state-of-the-art RL-based approaches and outperforms classical baselines in wall-clock synthesis time, improving both solution quality and runtime.

More broadly, this work highlights the value of fast, learned heuristics for scalable unitary synthesis. Unlike hand-crafted heuristics (e.g., those derived analytically in prior work such as Mosca & Mukhopadhyay (2021)), our heuristic is discovered automatically from data and can be deployed at scale with a standard search procedure. This combination—learned scoring plus efficient search—appears to be a practical path to improving synthesis performance on complex circuits where exhaustive or heavily structured methods become costly.

### ACKNOWLEDGMENTS

We thank Patrick Seifner for helpful discussions on this work. We also acknowledge the Lamarr Institute for Machine Learning and Artificial Intelligence as well as the University of Bonn for providing computational resources used in our experiments.

### FUNDING DISCLOSURE

This research has been funded by the Federal Ministry of Education and Research of Germany and the state of North Rhine-Westphalia as part of the Lamarr Institute for Machine Learning and Artificial Intelligence.

### AI-GENERATED CONTENT

We used GPT-5.2 (OpenAI) to assist with drafting and editing the manuscript text (wording, clarity and grammar). The authors produced and verified all technical content, claims, experiments and conclusions and take full responsibility for the final manuscript.

### REPRODUCIBILITY

To facilitate reproducibility, the main paper specifies the problem setup, residual-state formulation, method, training data construction, inference procedure, evaluation protocol, baselines and metrics in Sections 3–5. Appendix A provides additional model and algorithmic details for the transformer-based alternative, Appendix B reports training and hyperparameter details and Appendices C–D contain further experimental details and ablations. Code will be released publicly after the review/publication process.

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

## A   A TRANSFORMER-BASED ALTERNATIVE FOR THE MDL-PREDICTOR

While we used a simple MLP in our final configuration, we also investigated whether using a transformer-architecture Vaswani et al. (2017) as the MDL-predictor yields any improvements. This approach involved several steps:

### A.1   INPUT TOKENIZATION AND EMBEDDING

The model takes a unitary matrix $U \in \mathbb{C}^{2^n \times 2^n}$ as input. Each complex number in the matrix is treated as an individual token, which is processed through a multi-step procedure.

1. **Decomposition:** The input unitary $U$ is split into its real and imaginary components, forming two real-valued matrices, $U_{re}$ and $U_{im}$.

2. **Floating-Point Tokenization:** Each real number in both matrices is tokenized using a method similar to MDLformer Yu et al. (2025). A number is decomposed into a triplet of (sign, mantissa, exponent) based on a base-10 representation. For example, the value $-54.32$ would be tokenized into the triplet $(-, 5.432, \mathrm{E}+1)$. Each part of the triplet is then mapped to a discrete token from a shared vocabulary $\mathcal{V}$:

   - Sign: '+', '-'
   - Mantissa: e.g., 'N0000' to 'N9999' (for 4-digit precision)
   - Exponent: e.g., 'E-100' to 'E+100'

   This process converts each complex entry $z_{ij} = x_{ij} + iy_{ij}$ into a set of 6 discrete tokens (3 for the real part, 3 for the imaginary part).

   The 6 tokens for each matrix entry are converted to indices and embedded into dense vectors of dimension $d_{input}$. These 6 vectors are then concatenated and linearly projected to the model's main dimension, $d_{model}$, creating the final embedded representation grid $E \in \mathbb{R}^{2^n \times 2^n \times d_{model}}$.

$$v_{ij} = \mathrm{Concat}(\mathrm{Emb}(\mathrm{tokens}(x_{ij})), \mathrm{Emb}(\mathrm{tokens}(y_{ij}))) \in \mathbb{R}^{6 \times d_{input}} \tag{4}$$

$$E_{ij} = \mathrm{Linear}(\mathrm{Flatten}(v_{ij})) \in \mathbb{R}^{d_{model}} \tag{5}$$

3. **Alternative Tokenization in $\mathbb{Z}[\frac{1}{\sqrt{2}}, i]$:** It was shown in Giles & Selinger (2013) that the entries of the unitary matrices in the Clifford+T gate set can be expressed in the ring $\mathbb{Z}[\frac{1}{\sqrt{2}}, i]$. Instead of working with floating-point approximations, we can therefore express the matrix entries as follows:

$$z_{i,j} = \frac{1}{2^{k_1}}\left(a_{i,j} + b_{i,j} \cdot \sqrt{2}\right) + \frac{1}{2^{k_2}}\left((c_{i,j} + d_{i,j} \cdot \sqrt{2}) \cdot i\right) \tag{6}$$

   where $(a_{i,j}, b_{i,j}, c_{i,j}, d_{i,j}) \in \mathbb{Z}^4$ and $(k_1, k_2) \in \mathbb{N}^2$. We empirically found that the majority of elements can be bounded by $-\mathcal{Z} \le a_{i,j}, b_{i,j}, c_{i,j}, d_{i,j} \le \mathcal{Z}$ and $0 \le k_1, k_2 \le \mathcal{K}$. We therefore one-hot encode $a, b, c, d$ as vectors $v_{|a|} \in \{0,1\}^{\mathcal{Z}+1}$ and $v_{\mathrm{sign}(a)} \in \{0,1\}$ and $k$ as vectors $v_k \in \{0,1\}^{\mathcal{K}+1}$. We then embed these vectors using

$$d_{v_{|a|}} = \phi_{v_{|a|}}(v_{|a|}) \in \mathbb{R}^{d_{\mathrm{model}}}, \ d_{v_{\mathrm{sign}(a)}} = \phi_{v_{\mathrm{sign}(a)}}(v_{\mathrm{sign}(a)}) \in \mathbb{R}^{d_{\mathrm{model}}},$$
$$\text{and } d_k = \phi_k(v_k) \in \mathbb{R}^{d_{\mathrm{model}}}. \tag{7}$$

   Afterwards we compute

$$v_{i,j} = \mathrm{Linear}(\mathrm{Concat}(d_{k_1}, d_{v_{\mathrm{sign}(a)}}, d_{v_{|a|}}, d_{v_{\mathrm{sign}(b)}}, d_{v_{|b|}}, d_{k_2}, d_{v_{\mathrm{sign}(c)}}, d_{v_{|c|}},$$
$$d_{v_{\mathrm{sign}(d)}}, d_{v_{|d|}})) \in \mathbb{R}^{d_{\mathrm{model}}}. \tag{8}$$

   For the elements that fall outside of these bounds we use the learnable embedding $v_{\mathrm{NaN}} \in \mathbb{R}^{d_{\mathrm{model}}}$.

4. **2D Sinusoidal Positional Encoding:** To provide the model with spatial information, we use a static (non-learnable) 2D sinusoidal positional encoding, similar to the classical transformer Vaswani et al. (2017) $PE \in \mathbb{R}^{2^n \times 2^n \times d_{model}}$. This is added to the embedded representation.

$$H^{(0)} = E + PE \tag{9}$$

## A.2  Axial Attention Encoder

The core of the model is a stack of $L$ axial attention blocks. Each block consists of a standard Transformer Encoder Layer (which includes a multi-head self-attention mechanism followed by a position-wise feed-forward network) that is applied sequentially, first along the columns and then along the rows of the input grid. This allows the model to efficiently mix information across both spatial dimensions.

A single block transforms an input $H^{(l-1)}$ to $H^{(l)}$ as follows:

1. **Column-wise Processing:** The model first applies the Transformer Encoder Layer along the columns of the input tensor. This is achieved by reshaping the tensor so that the columns become the sequence dimension, applying the encoder layer, and then reshaping the result back to its original grid structure. This produces an intermediate representation, $H_{interim}$.

2. **Row-wise Processing:** The intermediate tensor $H_{interim}$ is then processed along its rows. This involves a second application of the same Transformer Encoder Layer, this time with the rows treated as the sequence dimension, to produce the final output of the block, $H^{(l)}$.

This sequence of operations is repeated for $L$ blocks.

## A.3  Pooling and Prediction Head

After the final encoder block, the resulting tensor $H^{(L)} \in \mathbb{R}^{2^n \times 2^n \times d_{model}}$ contains a rich, context-aware representation of the input unitary.

1. **Flattening:** The 2D grid of representations is flattened into a single sequence of length $N_{tokens} = (2^n)^2$.

$$H_{flat} = \text{Flatten}(H^{(L)}) \in \mathbb{R}^{N_{tokens} \times d_{model}} \tag{10}$$

2. **Attention Pooling:** To create a weighted summary of the token representations, we use attention pooling. A learnable query vector $w_{pool} \in \mathbb{R}^{d_{model}}$ computes attention scores.

$$\alpha_k = \frac{\exp(H_{flat,k} \cdot w_{pool})}{\sum_{j=1}^{N_{tokens}} \exp(H_{flat,j} \cdot w_{pool})} \tag{11}$$

$$H_{pooled} = \sum_{k=1}^{N_{tokens}} \alpha_k H_{flat,k} \quad \in \mathbb{R}^{d_{model}} \tag{12}$$

3. **Readout Head:** A final MLP takes the pooled vector and regresses it to a single scalar value, which is the predicted circuit gate count.

$$\widehat{CD} = \text{MLP}_{\text{out}}(H_{pooled}) \in \mathbb{R} \tag{13}$$

The entire architecture is trained end-to-end using a Mean Squared Error (MSE) loss between the predicted gate count $\widehat{CD}$ and the true optimal circuit gate count.

$$\mathcal{L} = (\widehat{CD} - CD_{true})^2 \tag{14}$$

## A.4  Comparison of Tokenization

We found that the more powerful transformer architecture does not yield noticeably better results and therefore does not justify the extra computational cost during inference and training (see Tab. 2).

Table 2: Comparison of three models on the validation set. Lower is better for Loss and MAE; higher is better for $R^2$.

| Model | Val. loss $\downarrow$ | MAE $\downarrow$ | $R^2 \uparrow$ |
|---|---|---|---|
| Float-tokenizer | 241.8 | 12.1 | **0.6** |
| Alternative-tokenizer | 251.0 | 12.3 | **0.6** |
| MLP | **67.1** | **7.7** | 0.55 |

## B  TRAINING AND HYPERPARAMETER DETAILS

**Baseline MLP (MDL predictor).**

- **Data generation.** We sample random Clifford+$T$ circuits using rejection sampling over $T$-counts, compute the corresponding unitaries, apply a greedy peephole optimizer and use the *optimized circuit gate count* as the regression target. For longer circuits, we add intermediate training examples by truncating the circuit after half and three-quarters of the $T$ gates (curriculum-style augmentation).
- **Architecture.** We train a 5-qubit MLP with hidden layer widths $\{1024, 512, 128\}$ and dropout 0.0.
- **Optimization.** We use AdamW ($\beta_1 = 0.9$, $\beta_2 = 0.999$, $\epsilon = 10^{-8}$) with mean-squared error (MSE) loss, weight decay 0.0, and gradient clipping at 1.0.
- **Training schedule.** The per-step batch size is 256 with 5-step gradient accumulation (effective batch size 1280). We train for 200 epochs with 2000 steps per epoch. We use a 2000-step warmup followed by cosine learning-rate decay with $T_{\max} = 20000$ and $\eta_{\min} = 10^{-6}$; the peak learning rate is $5 \times 10^{-4}$.
- **Data ranges and throughput.** Training covers gate counts in $[1, 60]$ and $T$-counts in $[0, 20]$. We use a replay buffer of size 6000 and a validation set of size 4096, with 29 dataloader workers and 200 validation steps per epoch. End-to-end training takes $\approx$6 hours on 30 CPU cores with a 4 GB GPU.

**CD-Former (transformer model).**

- **Inputs and target.** Unitaries are tokenized using the float tokenizer (mantissa precision 4; exponent range $[-10, 10]$) or the alternative tokenizer (see Sec. A). The supervision target is the *remaining* Clifford+$T$ gate count.
- **Architecture.** We use a 5-qubit Transformer encoder with model width 96, 4 attention heads, 8 encoder layers, feedforward dimension 384, dropout 0.1, and embedding dimension 96.
- **Optimization.** We use AdamW ($\beta_1 = 0.9$, $\beta_2 = 0.999$, $\epsilon = 10^{-8}$) with MSE loss, weight decay 0.01 and gradient clipping at 5.0.
- **Training schedule.** The per-step batch size is 240 with 5-step gradient accumulation (effective batch size 1200). We train for 100 epochs with 10k steps per epoch. We use a 50k-step warmup followed by cosine decay with $T_{\max} = 10^6$ and $\eta_{\min} = 10^{-5}$; the peak learning rate is $2 \times 10^{-4}$.
- **Data ranges and throughput.** Training covers gate counts in $[1, 60]$ and $T$-counts in $[0, 20]$. We use a replay buffer of size 4000 and a validation set of size 4096, with 29 dataloader workers and 200 validation steps per epoch.

## C  STRUCTURED CIRCUITS

This appendix collects the structured circuit families used throughout the experiments: GHZ states Greenberger et al. (1989), cluster states Raussendorf & Briegel (2001), phase gadgets as formalized in the ZX-calculus van de Wetering (2020), and the $[[5, 1, 3]]$ perfect code Laflamme et al. (1996). These circuits (shown in Fig. 4) serve as standardized targets with known structure, enabling controlled comparisons across qubit counts and circuit motifs.

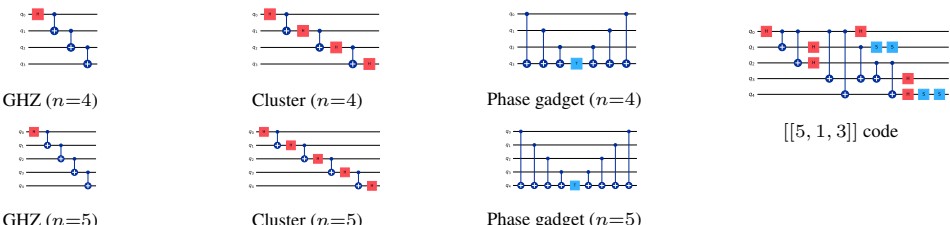

GHZ ($n=4$)  Cluster ($n=4$)  Phase gadget ($n=4$)  $[[5,1,3]]$ code

GHZ ($n=5$)  Cluster ($n=5$)  Phase gadget ($n=5$)

Figure 4: Canonical structured circuits used in our evaluation: GHZ states, cluster states, phase-gadget constructions and the $[[5,1,3]]$ (perfect) quantum error-correcting code.

## D  EFFECT OF EXPLORATION

A trial budget of 4,000 is the largest setting that fits comfortably on an 80 GB GPU using a $n = 5$ qubit model; 8,000 trials is exactly a $2\times$ increase. Accordingly, we implement an 8,000-trial evaluation by running two consecutive 4,000-trial calls, for a total runtime of $\sim$22 s. With two GPUs, these two 4,000-trial calls can be run in parallel, reducing the runtime to $\sim$11 s.

For this isolated exploration ablation we additionally train an $n = 4$ predictor to remove padding confounds; all main results use the single $n = 5$ predictor. To assess the effect of exploration, we vary the number of trials per circuit on the $n = 4$ qubit setting using an MLP trained on $n = 4$ qubits (i.e., no padding). Here 8000 trials can be run on a single GPU because the $n = 4$ model is smaller than the $n = 5$ model. For each time step $t \in \{0, \ldots, 20\}$ (same protocol as in Sec. 5), we measure the fraction of solved circuits and compare performance across trial budgets. Results are shown in Fig. 5.

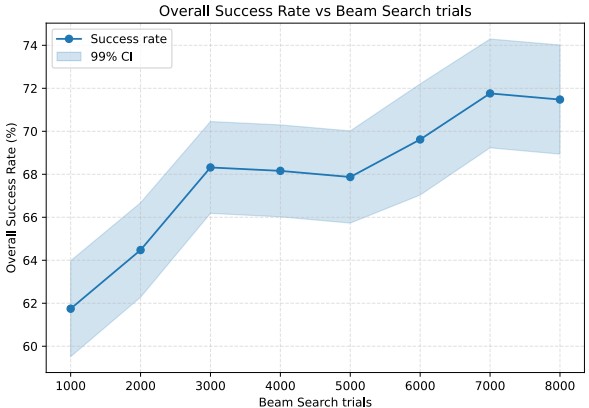

Figure 5: Overall success rate as a function of the beam-search trial budget.

Fig. 5 plots the mean overall success rate (solid line) as a function of the beam-search trial budget, aggregated over the evaluation protocol described above. Performance increases monotonically with additional trials, with the largest gains occurring when moving from low budgets (roughly 1,000–3,000 trials) and diminishing returns thereafter; beyond $\sim$5,000 trials, improvements are smaller and begin to plateau. The light-blue shaded region denotes a $99\%$ confidence interval around the mean success rate at each budget: narrower bands indicate more precise estimates, while wider bands indicate higher uncertainty.

This saturation is expected because the number of distinct symmetry transforms is bounded by $n!$ (and $\times 2$ if including inverses), so additional trials increasingly revisit previously explored transforms. Residual gains at higher budgets come primarily from stochasticity (Gumbel noise), which can introduce additional diversity even when sampling under the same symmetry.

