# OpenReview forum: "Towards Foundation Models for Quantum Unitary Synthesis via Zero-Shot MDL"
_ICLR.cc/2026/Workshop/FM4Science — ICLR 2026 Workshop FM4Science Poster_

### Official Review · Reviewer_9hY9 · 2026-02-19
**A Supervised Learning Approach to Quantum Unitary Synthesis via MDL-Guided Search**

**Rating:** 7
**Confidence:** 4

**Review:**

### Summary

This paper introduces an RL-free approach to quantum unitary synthesis by using a supervised MLP to approximate the Minimum Description Length (MDL) of residual unitaries. This model acts as a value function to guide a stochastic beam search. The authors demonstrate that this method achieves state-of-the-art success rates on Clifford+T synthesis while significantly reducing training time and offering zero-shot generalization across different qubit counts (up to $n=5$).

### Quality

The technical quality is high. The authors successfully address the "objective misalignment" problem in quantum synthesis by replacing standard numerical distance metrics with a structurally informed MDL value function. The experimental setup is thorough, comparing the proposed method against a diverse set of baselines, including classical heuristics (BFS/A*), exact solvers (QuantumCircuitOpt), and Reinforcement Learning models (AlphaZero-like). The use of the QAS-Bench suite for validation and the inclusion of budget-controlled wall-clock comparisons provides a fair and rigorous assessment of the algorithm's performance.

### Clarity

The paper is well-written and follows a logical progression from the problem statement to the final results. The explanation of the Markov Decision Process (MDP) over residual unitaries is clear, and the justification for the supervised learning approach (versus RL) is well-supported by the training time data. The figures are informative, particularly the heat map comparisons that visualize the success rate across varying circuit depths and T-counts.

### Originality

The originality lies in the "zero-shot" padding technique and the shift away from expensive RL-based search. While supervised learning for synthesis has been explored, the specific combination of MDL-guided beam search and the ability of a single model to generalize across qubit counts ($n=2$ to $n=5$) without retraining is a novel and practical contribution to the field of quantum compilation.

### Significance

This work is significant for the practical implementation of quantum algorithms on NISQ and early fault-tolerant hardware. The ability to synthesize deep Clifford+T circuits in minutes -- rather than the hours or days required by exact solvers or RL training -- makes this a viable tool for real-world quantum compiler pipelines. It bridges the gap between fast but suboptimal heuristics and computationally expensive optimal solvers.

### Pros

Training Efficiency: Small model size of MLP reduces training overhead compared to Reinforcement Learning baselines.
Generalization: Demonstrates true zero-shot capability across qubit counts via identity padding, removing the need for model-per-qubit training.
Search Performance: Outperforms traditional heuristics on "hard" circuits with high T-counts where greedy methods typically stall. Robustness: The use of Gumbel noise in the beam search provides a principled way to balance exploration and exploitation.

### Cons

Scalability Bottleneck: Because the method relies on dense matrix representations, it is inherently limited by memory costs to small-scale unitaries ($\le 5$ qubits).
Lack of Optimality Guarantee: As a heuristic search guided by an approximation, the method cannot guarantee the absolute shortest possible gate sequence.
Synthetic Bias: The performance is contingent on the synthetic training distribution, and it is unclear how the model handles unitaries drastically outside that distribution.

---

### Official Review · Reviewer_8vzZ · 2026-02-23
**Zero-Shot MDL**

**Rating:** 7
**Confidence:** 1

**Review:**

**Pros**

* This paper proposes a novel and highly efficient quantum unitary synthesis (QUS) method that combines supervised learning for Minimum Description Length (MDL) prediction with stochastic beam search, significantly reducing the training cost (from 7 days to about 6 hours) compared to state-of-the-art reinforcement learning baselines.
* The proposed identity-matrix zero-padding strategy enables impressive zero-shot generalization, allowing a single model trained on $n=5$ qubits to be directly deployed on tasks with fewer qubits without any retraining.
* The experimental evaluation is comprehensive and solid, clearly demonstrating the method's superior synthesis speed and success rate on both custom high T-count datasets and the standard QAS-Bench benchmark.
* The authors demonstrate a commendable and pragmatic research attitude by thoroughly comparing a complex Transformer architecture with a simple MLP in the appendix, ultimately adopting the MLP because it is more efficient and avoids the common pitfall of blindly stacking complex models.

**Cons**

* The method suffers from a fundamental scalability bottleneck because it relies on the explicit representation of dense unitary matrices, leading to a computational and memory overhead of $\Theta(4^n)$ that practically limits the approach to $n \le 5$ qubits.
* The "Minimum Description Length" (MDL) labels in the training data are merely approximations generated by a lightweight peephole optimizer rather than true global minimums, which may inherently cap the theoretical upper bound of the learned heuristic.
* The rigor of the baseline comparison is slightly compromised because the authors could not run the core reinforcement learning baseline (Rietsch et al., 2024) themselves, relying instead on reported numbers without strictly aligning hardware, memory, and time budgets.
* The use of the term "Foundation Models" in the title is highly exaggerated and misleading, given that the core model is actually just a lightweight multi-layer perceptron (MLP) with three hidden layers.

---

### Meta-Review · Area_Chair_Y7FN · 2026-02-27

**Recommendation:** Accept (Poster)
**Confidence:** 3

**Metareview:**

This paper provides a technically solid and practically impactful alternative to RL-based quantum synthesis, with substantial improvements in training efficiency, competitive benchmark performance, and a novel zero-shot qubit-scaling strategy that enhances deployment flexibility. Reviewers appreciate the thorough experimental evaluation and pragmatic architectural choices, but notice scalability limits due to dense matrix representations, reliance on approximate MDL labels, and imperfect baseline reproducibility alignment. Overall, this is a solid contribution to learned search heuristics for circuit synthesis.

---

### Decision · Program_Chairs · 2026-03-03

Accept (Poster)